# Association of Temporomandibular Disorder Symptoms with Physical Fitness among Finnish Conscripts

**DOI:** 10.3390/ijerph18063032

**Published:** 2021-03-16

**Authors:** Ossi Miettinen, Antti Kämppi, Tarja Tanner, Vuokko Anttonen, Pertti Patinen, Jari Päkkilä, Leo Tjäderhane, Kirsi Sipilä

**Affiliations:** 1Research Unit of Oral Health Sciences, University of Oulu, P.O. Box 5281, 90014 Oulu, Finland; tarja.tanner@oulu.fi (T.T.); vuokko.anttonen@oulu.fi (V.A.); kirsi.sipila@oulu.fi (K.S.); 2Department of Oral and Maxillofacial Diseases, University of Helsinki, P.O. Box 41, 00014 Helsinki, Finland; antti.kamppi@helsinki.fi (A.K.); leo.tjaderhane@helsinki.fi (L.T.); 3Medical Research Center Oulu (MRC Oulu), Oulu University Hospital, P.O. Box 5281, 90014 Oulu, Finland; 4Centre for Military Medicine, Finnish Defense Forces, P.O. Box 10, 11311 Riihimäki, Finland; pertti.patinen@fimnet.fi; 5Department of Mathematical Sciences, University of Oulu, P.O. Box 3000, 90014 Oulu, Finland; jari.pakkila@oulu.fi; 6Helsinki University Hospital, University of Helsinki, P.O. Box 41, 00014 Helsinki, Finland

**Keywords:** physical fitness, physical activity, endurance, BMI, TMD

## Abstract

Studies on the role of physical fitness, physical activity and obesity as risk factors for temporomandibular disorders (TMD) are scarce. The aim of the present study was to evaluate the association of TMD symptoms with physical fitness, physical activity and body mass index (BMI) among Finnish conscripts. The study sample consisted of 8685 Finnish conscripts. Data on self-reported TMD symptoms were used as outcome variables. Physical activity (questionnaire), physical fitness (measured with physical tests: Cooper test, push-ups, sit-ups and standing long jump) and body mass index (BMI) were used as explanatory variables. The associations between TMD symptoms and explanatory variables were evaluated using Chi-squared test and logistic regression analysis. The prevalence of all TMD symptoms was significantly higher among those who exercised more rarely than weekly. Pain-related TMD symptoms were also significantly more frequent among those who were overweight (BMI ≥ 25). Poor push-up results and overweight (BMI ≥ 25) were significantly associated with jaw pain and TMJ pain at jaw rest. The present study showed that good physical fitness may be a protective factor against TMD pain. Dentists should also be prepared to motivate TMD patients to physical activity and regular exercise as part of the treatment.

## 1. Introduction

Temporomandibular disorders (TMD) are musculoskeletal pain disorders of the masticatory system, i.e., the temporomandibular joints (TMJs) and the masticatory muscles [1]. The most common signs and symptoms of TMD are pain, limited range of jaw movement, and TMJ sounds. Clinical TMD diagnoses are based on TMD symptoms and clinical examination, based on the international, valid Diagnostic Criteria for TMD (DC/TMD) [2]. The prevalence of TMD symptoms and signs is quite common in the adult population, the levels showing large variation between studies. It has been shown that 15–50% of the population report at least one symptom associated with TMD, while 30–90% report at least one clinical sign [3,4]. TMD signs and symptoms are common in young adults and adolescents, the prevalence varying from 7 to 34% [5,6]. Our previous study evaluated the prevalence of TMD symptoms and their associations with alcohol consumption and smoking habits among Finnish conscripts. According to that study, the prevalence of different self-reported, at least occasional TMD symptoms varied between 5.8% (difficulty in jaw opening) and 27.8% (TMJ clicking) in the male population [7]. The etiology of TMD is considered complex and multifactorial, including occlusal disturbances, traumas, emotional stress, deep pain input and parafunctional activities, such as bruxism [8]. The role of different factors in the background of TMD still remains unclear.

The role of factors related to health behavior and lifestyle has recently been in the focus in chronic pain [9] as well as in TMD [10,11,12]. Our previous study showed that smoking was significantly associated with TMD symptoms except TMJ clicking. Additionally, consumption of alcohol at least once a week was significantly associated with facial pain, TMJ pain, TMJ pain at jaw rest, TMJ pain on jaw movement, and TMJ clicking. Use of snuff was significantly associated with facial pain [7]. Obesity is currently prevalent and can be associated with chronic pain, especially musculoskeletal pain [13]. A study concerning the association between body mass index (BMI) and TMD showed that non-obesity associated with TMD among females [14], whereas in another study on adolescents, no association was found between TMD-pain and BMI [13].

A study by Wänman [15] investigated endurance to physical strain in patients with TMD, showing that they have an impaired capacity to endure physically demanding tasks that specifically involve the jaw and shoulder girdle muscles. As TMD has been classified as a musculoskeletal disorder, it has been shown to have similar risk factors [16]. A cross-sectional study by Vierola et al. [16] investigated the associations of sedentary behavior, physical activity, cardiorespiratory fitness, and body fat content with pain conditions in prepubertal children. They found that physical activity was not associated with pain conditions, whereas high levels of sedentary behavior, low levels of cardiorespiratory fitness, and low body fat content had increased likelihood of various pain conditions. Physical activity, such as sports, has been found to be associated with musculoskeletal pains [17]. Auvinen et al. [17] examined the association of physical activity and the amount of sitting with low back pain (LBP). The study population consisted of 5999 members of the Northern Finland 1986 Birth Cohort at the age of 15–16 years. They found very active participation in physical activities in both sexes and a high amount of sitting in girls to be related to self-reported LBP. As the studies concerning the role of physical activity and fitness in TMD are scarce, further studies are needed to investigate their relationship as well. The large, comprehensive study on Finnish conscripts has data on both the prevalence of TMD and physical tests, offering a unique opportunity for this purpose.

### 1.1. Hypothesis of the Study

The hypothesis was that subjects with TMD symptoms show a lower capacity in physical trials and physical activity and are more frequently overweight compared to those without TMD symptoms.

### 1.2. Aim of the Study

The aim of the present study was to evaluate the association of TMD symptoms with physical fitness, physical activity and body mass index (BMI) among Finnish conscripts.

## 2. Materials and Methods

The epidemiological cross-sectional study was carried out in 20 garrisons (out of a total of 24) of the Finnish Defense Forces in January and July 2011. Four garrisons were excluded from the study due to outsourcing of dental services. However, the excluded garrisons were small; the total number of conscripts serving in them was less than 400. The screening of the conscripts’ oral health was carried out as part of the obligatory general health inspection during the conscripts’ first week in the military service. The oral health of all conscripts in 15 garrisons and every 5th conscript in alphabetical order (random sample) in the five largest garrisons included in the study was screened.

Oral health of a total of 13,819 conscripts (13,564 males and 255 females, mean age 19.6 years) born in 1990, 1991 or 1992 was examined clinically. In connection with oral screening, the conscripts had an opportunity to answer a computer-based questionnaire developed at the University of Oulu, Finland [18] for investigating individual background factors and health behaviors. A total of 8699 conscripts (of whom 8552 were men) answered the questionnaire.

### 2.1. Outcome Variables

Self-reported facial pain and symptoms of TMD were used as outcome variables. The TMD symptoms were inquired using the following six questions: “Have you had pain or ache in the face during the last year?”, “Have you had pain or ache in the jaws during the last year?” and “Have you had symptoms in the area of jaw joint? (pain at jaw rest, pain on jaw movement, clickings, difficulties in mouth opening)”. The response options for all these questions were no/yes (“occasionally”, “fairly often” and “often or continuously”). The answers were dichotomized as no: “no” or “occasionally” and yes: “fairly often” or “often or continuously”.

### 2.2. Explanatory Variables

Physical fitness tests, physical activity and body mass index (BMI) were used as explanatory variables. Physical fitness was evaluated by four physical tests: Cooper test (meters in 12 min of running), push-ups (number in 60 s), sit-ups (number in 60 s) and standing long jump (meters). The tests were guided by personnel that had been educated for the protocol. The most important standardized factors were the timing and order of the tests, meals, warming, practicing or pilot-testing, instructions and encouraging of the subjects. The Cooper test was performed on a separate day than the muscle fitness tests. If all these tests were performed during 3–4 days, Cooper tests were done first and the other tests afterwards. Before Cooper test, warming up for 10–15 min was performed. The muscle fitness tests were performed in a gym, and the Cooper test on a sports field.

The outcomes of the physical fitness tests were dichotomized. Cooper’s test result was considered poor if it was 0–2799 m and standing long jump if the result was 0–2.20 m, while in push-ups and sit-ups, poor result was 0–37 and 0–42 repetitions, respectively. Other results were considered good. The cut-offs for the tests are based on the instructions for Finnish Defence Forces [19] and on international standards [20,21,22]. Current physical activity was inquired using the following questions: “How often do you exercise?” (not at all, twice a month, 1–2 times in a month, 1–2 times in a week, 3–4 times in a week, more than five times in a week) and physical activity during the past 6 months was inquired as follows: “How often have you done sports or exercised in a gym during the last 6 months?” (never, hardly ever, now and then, almost every day, every day). The options for current physical activity were dichotomized as “inactive” (seldom than weekly) and “active” (at least weekly). The options for physical activity during 6 months were dichotomized as “inactive” (never, hardly ever, now and then) and “active (almost every day, every day). These dichotomizations were based on international recommendations for physical activity [23,24]. Each participant’s body mass index (BMI) was calculated from height and weight (based on measurements). The results were dichotomized as BMI < 25 and BMI ≥ 25, based on the limits set by the World Health Organization [25].

### 2.3. Statistical Analysis

The associations between TMD symptoms and explanatory variables were evaluated using Chi-squared tests. Unconditional multivariable logistic regression analysis was used to assess the associations between outcome and explanatory variables. The odds ratios (OR) and 95% confidence intervals (CI) were calculated. Statistical significance was set at *p* < 0.05.

All analyses were executed and figures drawn using SPSS software (version 25.0, SPSS, Inc., Chicago, IL, USA) and R software (version 2.13.2 patched, a language and environment for statistical computing, R Foundation for Statistical Computing, Vienna, Austria, http://www.R-project.org (accessed date: 1 February 2021)).

### 2.4. Ethical Considerations

The subjects gave their informed consent for the use of the data, and only data from those who had given their consent were used in the study. For the analyses, IDs were excluded. The research plan was accepted by the Ethics Committee of the Northern Ostrobothnia Hospital District on 29 March 2010. The Center for Military Medicine and the Defense Staff gave permission for the study in June 2010 (AG14218/23 June 2010).

## 3. Results

The prevalences of all TMD symptoms were significantly higher among currently physically inactive compared to active conscripts. In addition, the prevalences of all pain-related TMD symptoms except facial pain were significantly higher among those who reported less physical activity in the last 6 months compared to active conscripts. The prevalences of all pain-related TMD symptoms were significantly higher among those who were overweight (BMI ≥ 25) compared to those who were of normal weight (BMI < 25) (Table 1).

Those with poor outcomes in physical fitness tests showed higher prevalence of TMD symptoms than those with good physical fitness. Participants with poor push-up results reported significantly more TMD symptoms, with the exception of difficulties in jaw opening. Those who had poor Cooper test results reported significantly more TMJ pain on jaw movement and TMJ clicking than those with good results. Compared to the rest, those with poor sit-up results had higher prevalences of TMJ pain at jaw rest and TMJ clicking. (Table 1).

Physical inactivity, overweight (BMI ≥ 25) and female gender were significantly associated with jaw pain and TMJ pain at jaw rest (Table 2).

Of the physical fitness tests measured, poor push-up results and overweight (BMI ≥ 25) were significantly associated with jaw pain and TMJ pain at jaw rest. Female gender was the strongest explanatory factor for jaw pain and TMJ pain at jaw rest (Table 3).

## 4. Discussion

The results of the present study showed that low self-reported physical activity and poor physical fitness associate with the presence of TMD symptoms, especially those related to pain. The explanation for these associations may be complex and may occur through different ways. For example, psychosocial factors may associate with both physical activity [26] and with pain experience and central sensitization [27]. However, the present study did not evaluate psychosocial factors, and additional investigations are needed to assess their potential role. Other factors, such as insufficient sleep, may affect pain experience and also decrease physical activity [28]. Furthermore, the association between low physical activity and chronic pain may be explained by neurobiological mechanisms, linked with the dopaminergic system [9]. Although this study population comprised only selected healthy young adults with no severe physical or social disability, the results showed that the prevalence of self-reported TMD is already relatively high among subjects in their early adulthood. All the TMD symptoms were associated with low physical activity, and especially with reported current activity. Of the TMD symptoms, TMJ pain in jaw opening was least associated with physical fitness and activity. One explanation could be that those symptoms are more likely to be related to TMJ pathology rather than general pain mechanism with regard to facial pain, for example.

Of the fitness tests, poor push-up results correlated significantly with nearly all TMD symptoms, whereas there was more variation in the associations with other tests. Poor results in most of the physical fitness tests (except standing long jump) seem to associate especially with TMD pain symptoms, but not with difficulties in mouth opening, which is a less frequent condition and more specifically related to TMJ pathology rather than general pain mechanisms. The explanation for these results may be that the neck and shoulder muscles are functionally linked with the masticatory system [28,29], and exercise involving them thus also has an effect on the masticatory structures, acting at least partially as a protective factor against TMD symptoms. Wänman found that the vast majority of TMD patients had frequent pain in the cervical, shoulder and low back regions [15]. As high co-morbidity between pain in other sites and TMD has also been shown in other studies [30,31], it is most likely that they share a common underlying mechanism.

The present study showed that high BMI associated with pain-related TMD symptoms. Body mass index is not a golden standard to measure the amount of body fat, but it is commonly used to measure overweight and obesity in large population-based studies. Studies on the association between obesity and TMD are scarce; one study has found that obesity did not associate with the presence of TMD pain in adolescents [32], whereas based on a study from adult Korean population, TMD was associated with decreased BMI and abdominal obesity [14], thus showing results that are contradictory to the present study. Overweight and obesity can cause low-grade inflammation [33], which may in turn be related to chronic pain and TMD [34]. It can be speculated that psychosocial factors may be a link between high BMI and TMD symptoms. However, this was not investigated here, and more studies are needed to clarify the connection. However, in the present study the effect of BMI still remained significant in the multivariate analysis, indicating its independent effect.

The study sample mainly comprised a male cohort born in the early 1990s. The strength of the present study is its uniquely large study population, representing well Finnish conscripts but not the general population. It should be noted that the sample was a selected group, comprising young, healthy adults that had recently joined the military service, and were assumed to have good level of physical fitness. The number of females was low because they enter military service on a voluntary basis, and thus the female group can be regarded as even more selected than males were. Despite this, female gender showed a strong association with TMD symptoms, which is in line with a previous study [7]. The strength of the study was also that physical fitness was determined with four physical tests: Cooper test, push-ups, sit-ups and standing long jump. Cooper test represents aerobic fitness while push-ups, sit-ups and standing long jump are more representative of anaerobic muscular fitness.

An obvious limitation of the present study was that the data of TMD symptoms were based exclusively on self-report due to large sample size. These questions had not been validated to assess TMD. Thereby, the self-reported pain symptoms did not necessarily only represent TMD symptoms. Other possible causes can be infections related to erupting wisdom teeth and other dental pathologies. Referred pain is also one possible cause, as dental pain can be sensed in the masticatory structures. Nonetheless, the Northern Finland Birth Cohort 1966 (NFBC 1966) study found that facial pain based on these questions associated strongly with clinically assessed TMD [35]. Additionally, this cross-sectional study did not allow any estimates about the causal relationships. Another limitation of the study was the assessment of the physical activity, which was based on two, non-validated questions. Option “now and then” for the last 6 months of physical activity suggests occasional, not regular activity and the next option “almost every day” suggests activity more frequent than 3 times per week. This division may omit regular physical activity undertaken less than “almost every day” like 4 or 3 times per week. Therefore, the dichotomic division may cause underestimation in the physical active group.

All conscripts were evaluated to cope with demanding physical and mental challenges, with no severe diseases or conditions. Thus, the study population consisted of young healthy adults and does not represent the young Finnish adult population on general, although it represented 80% of those entering the military service. Annually 80% of each male cohort enters military service [36]. It has been shown that TMD conditions correlates with general diseases [37]. Thus, it can be speculated that TMD symptoms are even more common among young adults from the general population than the numbers presented in this study. In addition, about 1700 males complete non-military service annually for ethical reasons, and there are no data available on their physical or TMD conditions. As the proportion of female conscripts (n = 147) was small, generalizations must be done with caution here.

Self-managements are commonly used for the treatment of TMD, and the effectiveness of muscular exercise therapies for TMD has been shown in multiple studies [38,39,40,41]. Nonetheless, these studies have focused mainly on the effectiveness of jaw exercises in the TMD treatment, whereas the present study showed that good upper body muscle fitness had an inverse association with TMD symptoms and thus might act as a protective factor against TMD. It could be profitable to include upper body exercises in the self-management programs and therapeutic treatment of TMD, although further clinical trials are needed to investigate their effect.

## 5. Conclusions

In the treatment of TMD, the importance of overall muscle condition as a risk factor has received very little attention in previous studies. Based on this study, good physical fitness could at least partially protect from TMD pain. According to the study, it could be worthwhile to motivate the patient to lead a more physically active lifestyle as part of TMD treatment.

## Figures and Tables

**Table 1 ijerph-18-03032-t001:** Prevalence of TMD symptoms by gender, BMI, self-reported physical activity and physical fitness among Finnish conscripts.

		TMD Symptom (%)					
	n	Jaw Pain	TMJ Pain at Jaw Rest	Facial Pain	TMJ Pain on Jaw Movement	TMJ Clicking	Difficulties in Jaw Opening
Gender							
Men	8530	25.3	7.3	13.6	13.4	27.8	5.8
Female	147	33.3	15.0	14.3	18.4	25.9	12.9
*p*-value		0.027	0.000	0.806	0.079	0.610	0.000
95% C.I.		(0.83, 16.01)	(2.79, 14.38)	(−4.11, 7.31)	(−0.48, 12.08)	(−5.79, 8.38)	(2.58, 13.49)
BMI							
<25	6568	24.4	6.9	13.0	13.0	27.5	5.8
≥25	2109	28.6	8.9	15.6	15.2	28.7	6.3
*p*-value		0.000	0.002	0.003	0.010	0.285	0.397
95% C.I.		(2,04, 6.42)	(0.69, 3.42)	(0.90, 4.40)	(0.51, 3.98)	(−0.98, 3.44)	(−0.63, 1.74)
Physical activity in last 6 m							
Active	3565	23.5	6.3	12.8	12.3	25.4	5.4
Inactive	5112	26.8	8.2	14.2	14.4	29.5	6.3
*p*-value		0.001	0.001	0.061	0.005	0.000	0.081
95% C.I.		(1.44, 5.14)	(0.79, 2.99)	(−0.07, 2.84)	(0.64, 3.53)	(2.19, 5.99)	(−0.11, 1.89)
Current physical activity							
Active	6816	24.5	6.7	13.0	12.7	26.4	5.6
Inactive	1861	29.0	9.8	15.9	16.6	33.1	7.3
*p*-value		0.000	0.000	0.001	0.000	0.000	0.006
95% C.I.		(2.23, 6.83)	(1.69, 4.64)	(1.11, 4.80)	(2.09, 5.82)	(4.35, 9.1)	(0.46, 3.07)
Cooper test							
Good	1436	22.9	5.8	12.5	10.2	22.8	4.7
Poor	5497	25.3	7.3	13.6	13.2	28.6	5.9
*p*-value		0.061	0.047	0.275	0.002	0.000	0.080
95% C.I.		(−0.11, 4.80)	(0.15, 2.81)	(−0.91, 2.96)	(1.12, 4.73)	(3.27, 8.22)	(−0.16, 2.38)
Push-ups							
Good	2285	22.5	5.6	11.9	11.2	24.9	5.5
Poor	4908	25.9	7.8	14.0	13.2	28.8	5.8
*p*-value		0.002	0.001	0.015	0.017	0.006	0.610
95% C.I.		(1.27, 5.50)	(0.95, 3.37)	(0.42, 3.71)	(0.36, 3.57)	(1.70, 6.05)	(−0.89, 1.40)
Sit-ups							
Good	2087	23.1	5.7	12.6	12.1	25.2	4.7
Poor	5102	25.5	7.6	13.7	12.8	28.6	6.1
*p*-value		0.033	0.004	0.214	0.417	0.003	0.020
95% C.I.		(0.20, 4.54)	(0.62, 3.09)	(−0.65, 2.77)	(−1.02, 2.33)	(1.13, 5.61)	(0.23, 2.50)
Standing long jump							
Good	3682	23.6	6.4	12.7	12.1	27.1	5.4
Poor	3497	26.1	7.7	14.0	13.2	28.0	6.0
*p*-value		0.014	0.031	0.105	0.161	0.394	0.273
95% C.I.		(0.50, 4.50)	(0.12, 2.49)	(−0.27, 2.89)	(−0.44, 2.64)	(−1.17, 2.97)	(−0.47, 1.68)

**Table 2 ijerph-18-03032-t002:** OR and 95% confidence intervals (95% C.I.) from regression model on the association between jaw pain and TMJ pain at jaw rest and current physical activity and during the last six months.

	Jaw Pain			TMJ Pain at Jaw Rest		
	OR	95% C.I.		OR	95% C.I.	
		Lower	Upper		Lower	Upper
Gender						
Male	1			1		
Female	1.51	1.01	2.27	2.26	1.29	3.95
BMI						
<25	1			1		
≥25	1.15	1.03	1.30	1.23	1.01	1.49
Physical activity in last 6 m						
Active	1			1		
Inactive	1.15	1.01	1.31	1.33	1.07	1.65
Current physical activity?						
Active	1			1		
Inactive	1.22	1.05	1.41	1.28	1.01	1.62

**Table 3 ijerph-18-03032-t003:** OR and 95% confidence intervals (95% C.I.) from regression model on the association between jaw pain and TMJ pain at jaw rest and objectively measured physical performance results.

	Jaw Pain			TMJ Pain at Jaw Rest		
	OR	95% C.I.		OR	95% C.I.	
		Lower	Upper		Lower	Upper
Gender						
Male	1			1		
Female	1.46	0.90	2.38	2.59	1.37	4.88
BMI						
<25	1			1		
≥25	1.19	1.03	1.37	1.33	1.05	1.68
Cooper test						
Good	1			1		
Poor	0.98	0.82	1.19	1.06	0.76	1.47
Push-ups						
Good	1			1		
Poor	1.19	1.01	1.40	1.32	0.99	1.75
Sit-ups						
Good	1			1		
Poor	1.04	0.88	1.24	1.20	0.90	1.62
Standing long jump						
Good	1			1		
Poor	0.99	0.86	1.15	0.83	0.65	1.06

## Data Availability

The data presented in this study are available on request from the the corresponding author. The data are not publicly available due to ethical restrictions.

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
