# Peer review of "Association of Temporomandibular Disorder Symptoms with Physical Fitness among Finnish Conscripts"

_ijerph, 2021, doi:10.3390/ijerph18063032_

Round 1
Reviewer 1 Report
Firstly, I thank the Editorial Committee for the opportunity to review this manuscript. The authors present a relevant study, contributing to a better understanding of the association of temporomandibular disorder symptoms with physical fitness. Furthermore, the proposed manuscript meets adequately the purposes of the journal. Although the study includes a large sample, the methodological rigour must be improved.
Some recommendations are suggested below in order to improve the quality and methodological rigour of the manuscript for its publication.
The Introduction section is adequately carried out, including 18 references. The study aims are well stated. However, it is advisable to include a statement about how many signs and symptoms are needed to diagnose the temporomandibular disorder according to the international guidelines since the presence of a single sign/symptom does not imply the occurrence of the disorder.
The Material and Methods section should be deeply improved. The authors propose the publication of their research 10 years after collecting the data. As they stated in the Introduction section, signs and symptoms of this disorder are most prevalent in subjects between 20 and 40 years, but the mean age of the sample recruited is lower. Although the sample size is large, all participants are mainly males (only 1,7% are women) and they are all conscripts, not representing the general population. In this sense, it should be advisable not include female conscripts in the study, owing to the gender variable are unbalanced. Regarding outcome variables, the questionnaire of the self-reported facial pain and symptoms of this disorder is not validated. Was a pilot test carried out previously? On what were the questions based? The response options of this questionnaire do not include “no”, why? Why the response options were dichotomized in this way? In addition, it should be clarified if the presence of a single sign/symptom implies the occurrence of the disorder. In respect of explanatory variables, the questions about the current physical activity and the physical activity during the past 6 months are not validated and the reason for its dichotomization should be explained. In this sense, the dichotomization of the body mass index should be based on the current evidence. In conclusion, a better description and a validated measure of the outcome and explanatory variables is needed.
In general, the Results section is based on the obtained results after applying the research questionnaires and physical tests. However, in Table 1 the sample size is different depending on the variable measured or evaluated. How is possible this different sample size? This variation should be clarified and justified since the sample size should be the same in all measured variables. In addition, the results related to female gender are controversial, taking into account its low representativeness.
The Discussion and Conclusions sections are adequately carried out according to the results obtained by the authors. However, it is recommended to include in this section other relevant issues of the study, such as the representativeness of the female gender, the population recruited (they are all conscripts, young health adults newly joined the military service with an assumed good level of physical fitness and physical activity), its representativeness of the general population and the invalidated questions applied. All these issues should be also included as limitations of the study.
Finally, I consider the references should be updated (40.5% of the references belong to the last 5 years, 62.2% to the last 10 years, and, consequently, 37.8% are over 10 years).
Author Response
Firstly, I thank the Editorial Committee for the opportunity to review this manuscript. The authors present a relevant study, contributing to a better understanding of the association of temporomandibular disorder symptoms with physical fitness. Furthermore, the proposed manuscript meets adequately the purposes of the journal. Although the study includes a large sample, the methodological rigour must be improved.
Some recommendations are suggested below in order to improve the quality and methodological rigour of the manuscript for its publication.
The Introduction section is adequately carried out, including 18 references. The study aims are well stated. However, it is advisable to include a statement about how many signs and symptoms are needed to diagnose the temporomandibular disorder according to the international guidelines since the presence of a single sign/symptom does not imply the occurrence of the disorder.
Response: We thank the referee for the constructive comments. The diagnostics of TMD has been added in the Introduction (page 1, lines 39-41): “Clinical TMD diagnoses are based on TMD symptoms and clinical examination, based on the international, valid Diagnostic Criteria for TMD (DC/TMD) (Schiffman et al 2014)“, and the reference has been added in the reference list.
The Material and Methods section should be deeply improved. The authors propose the publication of their research 10 years after collecting the data. As they stated in the Introduction section, signs and symptoms of this disorder are most prevalent in subjects between 20 and 40 years, but the mean age of the sample recruited is lower.
Response: On the prevalence of TMD among adolescents, we added the following sentence in the Introduction (page 1, lines 44-46): “TMD signs and symptoms are common in young adults and adolescents, the prevalence varying from 7 to 34% [5,6]”.
Although the sample size is large, all participants are mainly males (only 1,7% are women) and they are all conscripts, not representing the general population. In this sense, it should be advisable not include female conscripts in the study, owing to the gender variable are unbalanced.
Response: We agree that the proportion of women was low. However, the aim was to collect a sample that represents well the Finnish conscripts, not only men, as women also have the opportunity to the service. The effect of gender is considered in the logistic regression analysis.
Regarding outcome variables, the questionnaire of the self-reported facial pain and symptoms of this disorder is not validated. Was a pilot test carried out previously? On what were the questions based? The response options of this questionnaire do not include “no”, why? Why the response options were dichotomized in this way? In addition, it should be clarified if the presence of a single sign/symptom implies the occurrence of the disorder
Response: Unfortunately the questions on facial pain and TMD symptoms are not validated. The original reason for using these questions in the present study was that they are same as the questions used in the Northern Finland Birth Cohort (NFBC) 1966 study (n=5696) (Rauhala et al 2000). Using the same questions enabled comparisons with Finnish general population, that has been reported in our earlier study on these conscripts (Miettinen et al 2017). We have also reported that these questions correlated strongly with clinically assessed TMD, based on a NFBC subsample that was examined clinically (Sipilä et al 2002), so we can convince that these questions have been tested.
The response option “no” was used, we apologize for this mistake and revised the text as followed (lines 133-1349: “The response options for all these questions were no/yes (“occasionally”, “fairly often” and “often or continuously”). The answers were dichotomized as no: “no” or“ occasionally” and yes: “fairly often” or “often or continuously”.
The responses were dichotomized in this way, because we though that occasional pain might not be clinically relevant, so selecting the “more- often” cases might reveal the more relevant TMD cases.
The critical discussion of the questionnaire is stated in the Discussion (lines 280-335): “ Obvious limitation of the present study was that the data of TMD symptoms were based exclusively on self-report due to large sample size. These questions had not been validated to assess TMD. Thereby, the self-reported pain symptoms did not necessarily only represent TMD symptoms. Other possible causes can be infections related to erupting wisdom teeth and other dental pathologies. Referred pain is also one possible cause, as dental pain can be sensed in the masticatory structures. Nonetheless, the Northern Finland Birth Cohort 1966 (NFBC 1966) study found that facial pain based on these questions associated strongly with clinically assessed TMD [32].”
In respect of explanatory variables, the questions about the current physical activity and the physical activity during the past 6 months are not validated and the reason for its dichotomization should be explained. In this sense, the dichotomization of the body mass index should be based on the current evidence. In conclusion, a better description and a validated measure of the outcome and explanatory variables is needed.
Response: We are aware about the limitations linked with physical activity assessment. However, due to the time limitations and the large study sample, any longer and more validated questionnaire could not be included in the study.
The international recommendation for physical activity for adults is at minimum at least 2.5 hours per week (http:// www.who.int/dietphysicalactivity/en/, www.health.gov/paguidelines). Based on these, we made the simple dichotomization for current PA “seldom that weekly”/”at least weekly”. Based on these, we made the dichotomization “seldom that weekly”/”at least weekly”. The reference of WHO guidelines has been added in the reference list. We apologize that there was a mistake in the dichotomization of 6 month-activity, this was revised (lines 152-153): “The options for physical activity during 6 months were dichotomized as “inactive” (never, hardly ever, now and then) and “active (almost every day, every day)” We added these references to the list and wrote the following sentence (lines 153-154): “These dichotomizations were based on international recommendations for physical activity [20,21].
The following sentence was added in the Discussion (lines 334-340): ”Another limitation of the study was the assessment of the physical activity, which was based on two, non-validated questions. Option “now and then” for last 6 months physical activity suggests occasional, not regular activity and the next option “almost every day” suggests activity more frequent than 3 times per week. This division may omit regular physical activity undertaken less than “almost every day” like 4 or 3 times per week. Therefore, the dichotomic division may cause underestimation in the physical active group.”
The limit of overweight was based on the limits set by WHO; commonly accepted BMI ranges are underweight (under 18.5 kg/m2), normal weight (18.5 to 25), overweight (25 to 30), and obese (over 30)(WHO Mean Body Mass Index (BMI)". World Health Organization. Retrieved 5 February 2019).
In general, the Results section is based on the obtained results after applying the research questionnaires and physical tests. However, in Table 1 the sample size is different depending on the variable measured or evaluated. How is possible this different sample size? This variation should be clarified and justified since the sample size should be the same in all measured variables. In addition, the results related to female gender are controversial, taking into account its low representativeness.
Response: A total of 8,699 conscripts (of whom 8,552 were men) answered the questionnaire. We apologize that in Table 1 the numbers for men/women and BMI classes were calculated from the total sample, not from those who responded to the questionnaire. The numbers have now been corrected. The slight variation in other numbers are due to missing values in questionnaire responses/ measurements. We agree that the proportion of women was low. However, the aim was to collect a sample that represents well all Finnish conscript, not only men, as women also have the opportunity to the service. The effect of gender was considered in the logistic regression analysis,
The Discussion and Conclusions sections are adequately carried out according to the results obtained by the authors. However, it is recommended to include in this section other relevant issues of the study, such as the representativeness of the female gender, the population recruited (they are all conscripts, young health adults newly joined the military service with an assumed good level of physical fitness and physical activity), its representativeness of the general population and the invalidated questions applied. All these issues should be also included as limitations of the study.
Response: The issues have now been discussed more (lines 270-276):”The study sample mainly comprised a male cohort born in the early 1990s. The strength of the present study is its uniquely large study population, representing well Finnish conscripts but not the general population. It should be noted that the sample was a selected, comprising young, healthy adults that had recently joined the military service, and were assumed to have good level of physical fitness. The number of females was low because they enter military service on a voluntary basis, and thus the female group can be regarded as even more selected than males were.”
and (lines 283-330): ”Obvious limitation of the present study was that the data of TMD symptoms were based exclusively on self-report due to large sample size. These questions had not been validated to assess TMD.
Finally, I consider the references should be updated (40.5% of the references belong to the last 5 years, 62.2% to the last 10 years, and, consequently, 37.8% are over 10 years).
Response: We agree with the referee and deleted several old references, and replaced them by newer ones.
Reviewer 2 Report
Dear author congratulations on your paper, researchers each day bring new benefit to exercise which I find very important. The introduction and the discussion are very nice presented, however the methodology and the results should be improved.
Here you have my comments:
Line 57. It is one study, no estudies
Line 83. Because the objetive of the study is not to compare genders, I would eliminate women of the results or you can add gender to the objective.
Line 99. “(19),” should be after Finland
Line 102. Women should be not included in the analyses or added to the objective.
Line: 111 why were the answers dichotomized? I would not dichotomized them so we can have more information.
Line 117. Why were the tests results dichotomized? I would not dichotomized them so we can have more information. What was the criterium to evaluate the fitness test results as poor or good?
Line 120. In “How often do you exercise? two answers were almost the same, twice a month and 1-2 times in a month. I do not understand that. Why didn´t the authors used a validated questionary such as “The International Physical Activity Questionnaire (IPAQ)”? Was the questionary used in this research validated?
It seems that the use two questions to ask for the physical activity, the second is in the last 6 months but the first does not say anything. In the results you can read that it seams that is current physical activity but you cannot read that in the question
I would eliminate the physical activity part off of this paper since it was not properly evaluated.
Line 126. How was the height and the weight measured? If it was self reported I should be written.
Line 127. It should be explained why 25 was the cut off point. However I also believe the author can do a correlation.
Line 128. Statistical analysis
I will not dichotomized the variables, and I would use a correlation instead a regression, it is easier to understand and provide more information.
Line 166. Should be confidence and not confident also in line 169.
Line 255
“Good physical fitness promotes overall well-being and physical health.” this is not a conclusión of this study.
The references need to be in the same format, for example some years are in bolt and some others no.
Author Response
Dear author congratulations on your paper, researchers each day bring new benefit to exercise which I find very important. The introduction and the discussion are very nice presented, however the methodology and the results should be improved.
Response: We thank the referee for the encouraging and constructive comments. We have now revised the manuscript based on these.
Here you have my comments:
Line 57. It is one study, no estudies
Response: thank you, this was corrected (line 66)
Line 83. Because the objetive of the study is not to compare genders, I would eliminate women of the results or you can add gender to the objective.
Response: We agree that the proportion of women was low. However, the aim was to collect a sample that represents well all Finnish conscript, not only men, as women also have the opportunity to the military service. The effect of gender is considered in the logistic regression analysis,
Line 99. “(19),” should be after Finland
Response: corrected (line 124).
Line 102. Women should be not included in the analyses or added to the objective.
Response: please see the previous comment.
Line: 111 why were the answers dichotomized? I would not dichotomized them so we can have more information.
Line 117. Why were the tests results dichotomized? I would not dichotomized them so we can have more information. What was the criterium to evaluate the fitness test results as poor or good?
Response: the dichotomizations grouped the sample for clinically relevant subsamples. Also for the regression analyses, the independent variables must dichotomous.
Line 120. In “How often do you exercise? two answers were almost the same, twice a month and 1-2 times in a month. I do not understand that. Why didn´t the authors used a validated questionary such as “The International Physical Activity Questionnaire (IPAQ)”? Was the questionary used in this research validated?
It seems that the use two questions to ask for the physical activity, the second is in the last 6 months but the first does not say anything. In the results you can read that it seams that is current physical activity but you cannot read that in the question
I would eliminate the physical activity part off of this paper since it was not properly evaluated.
Response: we agree with the referee that the IPAQ questionnaire has acceptable validity when assessing levels and patterns of PA in healthy adults). However, due to the time limitations and the large study sample, this questionnaire could not be included in the study. The international recommendation for physical activity for adults is at minimum at least 2.5 hours per week (http:// www.who.int/dietphysicalactivity/en/, www.health.gov/paguidelines). Based on these, we made the simple dichotomization for PA “seldom that weekly”/”at least weekly”. Despite the deficiencies, we decided to leave this variable, as it gives some information of the PA of the subject.
Line 126. How was the height and the weight measured? If it was self reported I should be written.
Response: These were based on measurements, not on self-report. This is clarified in the text (lines 155-156): “Each participant’s body mass index (BMI) was calculated from height and weight. (based on measurements)”
Line 127. It should be explained why 25 was the cut off point.
Response: The limit of overweight was based on the limits set by WHO; commonly accepted BMI ranges are underweight (under 18.5 kg/m2), normal weight (18.5 to 25), overweight (25 to 30), and obese (over 30), based on WHO "WHO Mean Body Mass Index (BMI)". World Health Organization. Retrieved 5 February 2019. We added these references to the list.
Line 128. Statistical analysis
I will not dichotomized the variables, and I would use a correlation instead a regression, it is easier to understand and provide more information.
Response: We prefer using regression, as it assesses the effect of multible variables. In statistical modeling, regression analysis is a set of statistical processes for estimating the relationships between a dependent variable (TMD symptom here) and one or more variables. The confounding effect of gender and BMI could be controlled with this method.
Line 166. Should be confidence and not confident also in line 169.
Response: we thank the referee for these attentions. The legends of Tables 2 and 3 has been revised accordingly.
Line 255
“Good physical fitness promotes overall well-being and physical health.” this is not a conclusión of this study.
Response: this sentence was deleted from the conclusion (lines 367-368)
The references need to be in the same format, for example some years are in bolt and some others no.
Response: we have checked the reference list and revised it.
Reviewer 3 Report
Thank you for the opportunity to review this well-written and designed paper. Authors assessed associations between temporomandibular disorder symptoms and physical activity (based on subjective questionnaire), physical fitness (based on aerobic and anaerobic fitness tests) and BMI among a large group of Finnish conscripts.
The main strength of present study beyond the large sample is good study design and sampling procedures. Authors clearly describe applied methods and obtained results, however I would like to share some of minor suggestions that authors could consider.
The main limitation of study, not mentioned by authors is the physical activity assessment based on only two questions, instead of at least short, validated physical activity questionnaire. Option “now and then” for last 6 months physical activity suggests occasional, not regular activity and the next option “almost every day” suggests activity more frequent than 3 times per week. This division seems to omit regular physical activity undertaken less than “almost every day” like 4 or 3 times per week which seems to be very common in the active population. Taking into consideration that between these two options authors set dichotomic division there could be considerable risk of underestimation in the physical active group which could lead to bias in further analysis.
Material and methods.
I would like to suggest that authors detailed in text exact sample sizes for particular explanatory variables since physical fitness and physical activity samples were not equal.
Line 101 – Please consider providing more detail on the time limit applied in questionnaire data collection
Lines 116-119 – I think that paper would benefit if the authors mention and cite the grounds on which they based the cut-off points for physical fitness assessment (2,799 and 2.2 meters for Cooper test and long jump respectively, 37 and 42 repetitions for push-ups and sit-ups respectively)
Lines 124-125 – I would suggest to reconsider the description of how physical activity answers were dichotomized. In present form the description suggests that both current and last 6 months physical activity were dichotomized to “seldom than weekly” and “at least weekly” which could be confusing since the options for physical activity in last 6 months do not include any reference to week. It will be much easier for the reader to follow if the authors stay in line with the descriptions in text and those in tables.
Table 2 and 3 – Please consider adding sample sizes in each table since it is not the same and without this information the reader could be confused with different OR and 95% C.I. values relating to the same variables (Gender, BMI)
Author Response
Thank you for the opportunity to review this well-written and designed paper. Authors assessed associations between temporomandibular disorder symptoms and physical activity (based on subjective questionnaire), physical fitness (based on aerobic and anaerobic fitness tests) and BMI among a large group of Finnish conscripts.
Response: We thank the referee for the encouraging and constructive comments. We have now revised the manuscript based on these.
The main strength of present study beyond the large sample is good study design and sampling procedures. Authors clearly describe applied methods and obtained results, however I would like to share some of minor suggestions that authors could consider.
The main limitation of study, not mentioned by authors is the physical activity assessment based on only two questions, instead of at least short, validated physical activity questionnaire. Option “now and then” for last 6 months physical activity suggests occasional, not regular activity and the next option “almost every day” suggests activity more frequent than 3 times per week. This division seems to omit regular physical activity undertaken less than “almost every day” like 4 or 3 times per week which seems to be very common in the active population. Taking into consideration that between these two options authors set dichotomic division there could be considerable risk of underestimation in the physical active group which could lead to bias in further analysis.
Response: We agree that the variables used for physical activity may cause bias. The validated questionnaire would have been more reliable, f ex the IPAQ questionnaire that has acceptable validity when assessing levels and patterns of PA in healthy adults. However, due to the time limitations and the large study sample, this questionnaire could not be included in the study. The international recommendation for physical activity for adults is at minimum at least 2.5 hours per week (http:// www.who.int/dietphysicalactivity/en/, www.health.gov/paguidelines). Based on these, we made the simple dichotomization for PA “seldom that weekly”/”at least weekly”. Despite the deficiencies, we decided to leave this variable, as it gives some information of the PA of the subject. We have now added the use of questions on PA as a limitation of the study (lines 336-342) as follows: “Another limitation of the study was the assessment of the physical activity, which was based on two, non-validated questions. Option “now and then” for last 6 months physical activity suggests occasional, not regular activity and the next option “almost every day” suggests activity more frequent than 3 times per week. This division may omit regular physical activity undertaken less than “almost every day” like 4 or 3 times per week. Therefore, the dichotomic division may cause underestimation in the physical active group.
.Material and methods.
I would like to suggest that authors detailed in text exact sample sizes for particular explanatory variables since physical fitness and physical activity samples were not equal.
Line 101 – Please consider providing more detail on the time limit applied in questionnaire data collection
Lines 116-119 – I think that paper would benefit if the authors mention and cite the grounds on which they based the cut-off points for physical fitness assessment (2,799 and 2.2 meters for Cooper test and long jump respectively, 37 and 42 repetitions for push-ups and sit-ups respectively)
Response: The cut-offs for the tests are based on the instructions for Finnish Defence Forces , based on Pihlainen et al 2011 (https://puolustusvoimat.fi/documents/1948673/2258811/PEVIESTOS-kuntotestaajank%C3%A4sikirja-2015/332148cf-be2e-49ea-8fa2-0df6423724fc). This was added in the methods (lines 144-145).
Lines 124-125 – I would suggest to reconsider the description of how physical activity answers were dichotomized. In present form the description suggests that both current and last 6 months physical activity were dichotomized to “seldom than weekly” and “at least weekly” which could be confusing since the options for physical activity in last 6 months do not include any reference to week. It will be much easier for the reader to follow if the authors stay in line with the descriptions in text and those in tables.
Response: We apologize for this confusion. Due to a flaw, we have now corrected the definition of dichotomization of 6-month-physical activity as follows (lines 152-153): “The options for physical activity during 6 months were dichotomized as “inactive” (never, hardly ever, now and then) and “active (almost every day, every day).” We have also added terms active/inactive in Table 2., uniformly with the other text.
Table 2 and 3 – Please consider adding sample sizes in each table since it is not the same and without this information the reader could be confused with different OR and 95% C.I. values relating to the same variables (Gender, BMI)
Response: The ORs and 95% CIs relating to the same variables are different, because separate models different variables were used. The sample sizes are presented in Table 1.
Round 2
Reviewer 1 Report
Firstly, I thank the Editorial Committee for the opportunity to review this manuscript again. I congratulate the authors for their great effort to increase the quality of the proposed manuscript, improving mainly its methodological rigour. They have followed all the recommendations and suggestions proposed. According to my recommendations, they have clarified perfectly all my questions and doubts, improving the manuscript's reading and facilitating its understanding for the reader. Furthermore, they have clarified some issues in the manuscript, including relevant phrases and sentences. Consequently, they have increased significantly the quality of the manuscript for its acceptance. Therefore, the manuscript could be considered as accepted for its publication.
Author Response
Firstly, I thank the Editorial Committee for the opportunity to review this manuscript again. I congratulate the authors for their great effort to increase the quality of the proposed manuscript, improving mainly its methodological rigour. They have followed all the recommendations and suggestions proposed. According to my recommendations, they have clarified perfectly all my questions and doubts, improving the manuscript's reading and facilitating its understanding for the reader. Furthermore, they have clarified some issues in the manuscript, including relevant phrases and sentences. Consequently, they have increased significantly the quality of the manuscript for its acceptance. Therefore, the manuscript could be considered as accepted for its publication.
Response: We thank the referee for encouraging comments.
Reviewer 2 Report
Dear authors.
The improvement in you paper is evident, congratulations. Although I believe that the statistics could be improved, however your way of presenting the data is very simple and easy to understand.
Here you have some comments about some aspects that you can still improve.
Lines 94-95, I would delete “, representing comprehensively males” because for me it is confusing is you want to represent the whole conscripts.
Line 145. Could you elaborate a little how they make the cut-off? Most of the people interested in this paper will not understand the Finnish Defence Forces book.
Line 145. What do you mean by current physical activity? In the last month? Usually current PA is measured in the last week, but since in your answers you can read per month, it is confusing for all of us familiar with this kind of questionaries. it is important to differentiate between current PA and PA in the last 6 months.
Line 155. How were the measurements made? Please, specify who made them and the equipment used.
Line 156. You explained on the comment that 25 was use because WHO you used 25 as a cut off point for overweight, although I believe this is common knowledge it will be interesting to see it on the paper. I like the way you explained it in lines 194 and 195, but it should be explained in the methodology.
Author Response
The improvement in you paper is evident, congratulations. Although I believe that the statistics could be improved, however your way of presenting the data is very simple and easy to understand.
Response: We thank the referee for encouraging comments.
Here you have some comments about some aspects that you can still improve.
Lines 94-95, I would delete “, representing comprehensively males” because for me it is confusing is you want to represent the whole conscripts.
Response: this was deleted (line 87)
Line 145. Could you elaborate a little how they make the cut-off? Most of the people interested in this paper will not understand the Finnish Defence Forces book.
Response: The following revision was made (line 130): “The cut-offs for the tests are based on the instructions for Finnish Defence Forces [19] and on international standards [20-22].” We added the following references from the Finnish Defence Forces book:
American College of Sports Medicine. ACSM´s guidelines for exercise testing and prescription. 6th ed. Philadelphia, Lippincott Williams & Wilkins. 2000.
Cooper, K. A means of assessing maximal oxygen intake. Correlation between field and treadmill testing. JAMA 1968; 203: 201-204.
Fletcher G., Balady G. & Blair S. Statement on exercise: benefits and recommendations for physical activity programs for all Americans: a statement for health professionals by the Committee on Exercise and Cardiac Rehabilitation of the Council on Clinical Cardiology, American Heart Association. Circulation 1996; 94:857–862.
Line 145. What do you mean by current physical activity? In the last month? Usually current PA is measured in the last week, but since in your answers you can read per month, it is confusing for all of us familiar with this kind of questionaries. it is important to differentiate between current PA and PA in the last 6 months.
Response: The questions concerning current physical activity aimed to clarify the present activity, whereas 6-month PA during a longer time period. Current physical activity was inquired using the following questions: “How often do you exercise?” (not at all, twice a month, 1–2 times in a month, 1–2 times in a week, 3–4 times in a week, more than five times in a week).
Line 155. How were the measurements made? Please, specify who made them and the equipment used.
Response: The following description of the tests were added (lines 119-125): “The tests were guided by personnel that had been educated for the protocol. The most important standardized factors were the timing and order of the tests, meals, warming, practicing or pilot-testing, instructions and encouraging of the subjects. The Cooper test was performed on a separate day than the muscle fitness tests. If all these tests were performed during 3-4 days, Cooper tests were done first and the other tests afterwards. Before Cooper test, warming up for 10-15 min was performed. The muscle fitness tests were performed in a gym, and the Cooper test on a sports field.”
Line 156. You explained on the comment that 25 was use because WHO you used 25 as a cut off point for overweight, although I believe this is common knowledge it will be interesting to see it on the paper. I like the way you explained it in lines 194 and 195, but it should be explained in the methodology.
Response: WHO limits have been added in Methods (lines 141-142): The results were dichotomized as BMI <25 and BMI ≥25, based on the limits set by the World Health Organization [25].